# Efficient rescue of damaged neural networks

## Abstract

Neural networks in the brain and in neuromorphic chips confer systems with the ability to perform multiple cognitive tasks. However, both kinds of networks experience a wide range of physical perturbations, ranging from damage to edges of the network to complete node deletions, that ultimately could lead to network failure. A critical question is to understand how the computational properties of neural networks change in response to node-damage and whether there exist strategies to repair these networks in order to compensate for performance degradation. Here, we study the damage-response characteristics of two classes of neural networks, namely multilayer perceptrons (MLPs) and convolutional neural networks (CNNs) trained to classify images from MNIST and CIFAR-10 datasets respectively. We also propose a new framework to discover efficient repair strategies to rescue damaged neural networks. The framework involves defining damage and repair operators for dynamically traversing the neural networks loss landscape, with the goal of mapping its salient geometric features. Using this strategy, we discover features that resemble path-connected attractor sets in the loss landscape. We also identify that a dynamic recovery scheme, where networks are constantly damaged and repaired, produces a group of networks resilient to damage as it can be quickly rescued. Broadly, our work shows that we can design fault-tolerant networks by applying on-line retraining consistently during damage for real-time applications in biology and machine learning.

## 1   Introduction

Living neural networks in the brain and artificial networks engineered on neuromorphic chips [1] perform an array of computational and information processing tasks [2, 3, 4]. However, both these networks are susceptible to physical perturbations that lead to a decline in functional performance [5]. Understanding how damage of neural units in a network leads to cognitive decline is of great interest to biomedical sciences as well as to AI practitioners implementing artificial networks on neuromorphic hardware. In addition, deciphering techniques to 'search' for neural networks that are resilient to perturbation and strategies that efficiently rescue damaged networks to compensate for performance degradation are of great interest to both the communities. So far, researchers have only focused on studying resilience of neural nets to perturbation of input signals [6] by generating adversarial examples [7] that highlight the vulnerability of neural nets. However, not much has been done towards understanding the decline in performance due to physical perturbation of neural networks [8] and unraveling repair strategies to rescue damaged networks.

In this paper, inspired by the powerful paradigms introduced by deep learning, we attempt to understand the computational and mathematical principles that impact the ability of neural networks to tolerate damage and be repaired. We characterize the response of two classes of neural networks, namely multilayer perceptrons (MLP's) and convolutional neural nets (CNN's) to node-damage and propose a *new framework* that identifies strategies to efficiently rescue damaged networks in a principled fashion.

Our key contribution is the introduction of a framework that conceptualizes damage and repair of networks as operators of a dynamical system in the high-dimensional parameter space of a neural network. The damage and repair operators are used to dynamically traverse the landscape with the goal of mapping local geometric features [9, 10] (like, fixed points, limit-cycles or point/line-attractors) of the neural networks' loss landscape. The framework led us to discovering that the iterative application of damage and repair operators results in networks that are highly resilient to node-deletions as well as guides us to uncover the presence of geometric features that resemble a path-connected attractors set, in many respects, in the neural networks' loss landscape. Attractor-like geometric features in the networks' loss landscape explains why the iterative damage-repair strategy always results in the rescue of damaged networks within a small number of training cycles.

## 2   Susceptibility of neural networks to damage

The first question we ask in this paper is how do neural networks respond to physical perturbations and how does it affect their functional performance. We characterize the impact of neural damage on 'cognitive' performance of neural networks by tracking the performance of two classes of artificial neural networks, namely MLPs and CNNs, to deletion of neural units from the network. The MLPs and CNNs were trained to perform simple cognitive tasks like image classification on MNIST and CIFAR-10 datasets respectively before the networks were perturbed.

Submitted to 33rd Conference on Neural Information Processing Systems (NeurIPS 2019). Do not distribute.

To damage a node $i$ in the hidden layer of an MLP or in the fully connected layer of a CNN, we zero all connections between node $i$ and the rest of the network. And, to damage a node $j$ in the convolutional layer of a CNN, we zero the entire feature map. In this paper, we are specifically interested in node-damage as our perturbation because of its similarity in phenomena to neuron death in biological networks and node-failures in neuromorphic hardware.

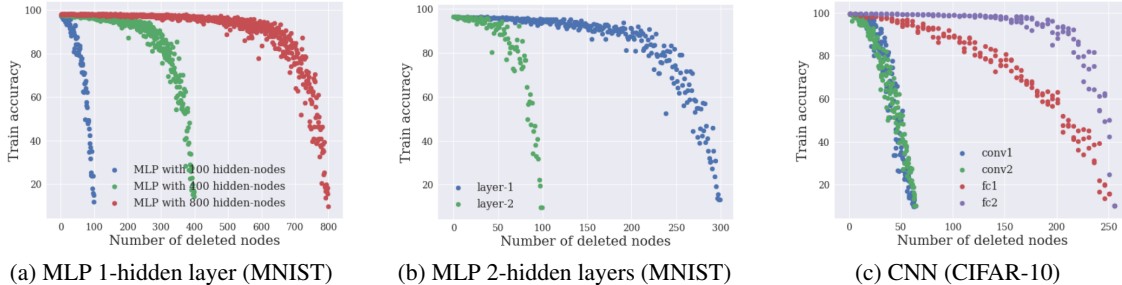

(a) MLP 1-hidden layer (MNIST)  (b) MLP 2-hidden layers (MNIST)  (c) CNN (CIFAR-10)

Figure 1: **Damage of neural units in artificial neural networks (Phase transition)** (a) Performance of MLP with 1 hidden layer in MNIST classification (b) Performance of MLP with 2 hidden layers in MNIST classification (c) Performance of CNN in CIFAR-10 classification

We observe a steep increase in the rate of decline of functional performance as we incrementally delete nodes from either an MLP with 1 hidden layer (Fig-1a), an MLP with 2 hidden layers (Fig-1b) or a CNN with 2 convolutional layers, a pooling layer and 2 fully connected layers (Fig-1c). We refer to this discrete jump in the rate of decline of performance as a **phase transition.**

The existence of a phase transition shows that neural nets (MLP's and CNN's) damaged above their respective critical thresholds are not resilient to any further perturbation. We are interested in deciphering strategies that enable the quick rescue of damaged neural nets and also want to identify networks that are more resilient to perturbation.

## 3   Can we rescue these damaged networks?

We ask whether it is fundamentally possible to rescue damaged networks in order to compensate for their performance degradation. To do so, we re-train damaged networks via two strategies mentioned below: **(Strategy-1: Functional sub-network retraining)** Purely re-train the 'functioning' sub-network, ie the weights connecting damaged nodes are kept at zero, while enabling plasticity for weights connecting the remaining undamaged nodes. **(Strategy-2: Node replacement)** Replace the damaged units with 'embryonic' nodes and retrain the network such that 'embryonic' nodes are more plastic than the nodes in the functioning sub-network.

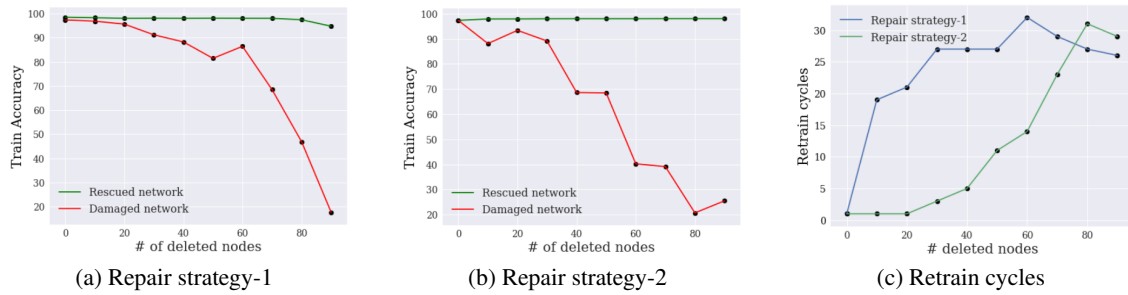

(a) Repair strategy-1  (b) Repair strategy-2  (c) Retrain cycles

Figure 2: **Repairing damaged networks** (a,b) The red line is the accuracy of the network after $m$ nodes have been damaged. The green line is the accuracy of network after it has been retrained using one of two strategies. (a) [Strategy-1] Purely retraining the functioning sub-network. (b) [Strategy-2] Replacing damaged nodes with 'embryonic' nodes and selectively retraining only the newly replaced nodes. (c) The number of training cycles required for repairing a damaged network with $m$ nodes via both strategies.

The plots in figure-2 show that damaged neural networks can be rescued to regain their original functional performance when re-trained via both strategies 1 and 2. However, they require a large number of training cycles (epochs) to be effectively rescued (figure-2c). The requirement of a large number of training cycles for the effective rescue of a neural network *reduces the feasibility* of either strategy as it isn't ideal for both, living neural networks in the brain or artificial networks implemented on neuromorphic hardware to be re-trained for extended periods of time to recover from small damages to its network.

## 4   Iterative repair of networks v/s batch repair of networks

Inspired by the dynamic recovery paradigm adopted by most biological systems, where networks are constantly being perturbed and repaired, we propose an iterative damage-repair strategy and test whether this produces networks that are more resilient to perturbation as well as if it allows us to rescue damaged networks with much lesser training cycles.

66  Figure-3 demonstrates that the iterative damage-repair paradigm can rescue neural networks to their
67  original functional performance within 15 training cycles! This is in stark contrast to the batch recovery
68  of networks, either via strategy 1 or 2, as they need up to 35 training cycles to repair networks with small
69  damages. It is important to note that although iterative-rescue constantly damages and repairs networks,
70  the repair operation doesn't revive any of the damaged nodes, ie the damaged nodes and its weights
71  remain 0. We stress that the constant perturbation and repair strategy allows us to reach a favorable
72  'space' in the networks' loss manifold that contains high performing, more resilient, sparser networks.

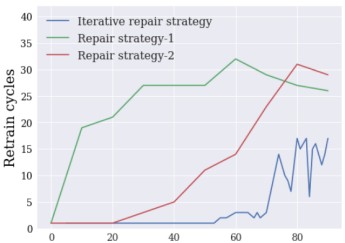

73  As the iterative process of damage and repair always enabled the fast recovery of a damaged network
74  (irrespective of the number of damaged units), this was surprising to us and we were interested in
75  determining if the loss landscape manifold had 'special' geometric features that enabled this rescue.

Figure 3: Iterative damage-repair strategy enables swift recovery of the network, when compared to batch-recovery of the network achieved by either strategy 1 or 2.

76  To map geometric features of a neural networks' loss landscape, we formally conceptualize the iterative
77  damage-repair paradigm as a dynamical system that involves the application of a damage and repair
78  operator ($r$) on a neural network ($w$).

79  We define $w$ to be a feed-forward neural network with $n$ nodes and $N$ total connections.

$$w = [\vec{w_1}, \vec{w_2}, ..., \vec{w_i}, ..., \vec{w_n}]$$

80  Here, $\vec{w_i}$ is the set of connections made by node $i$ with the previous layer in the network. By definition, $\vec{w_i} = \phi$, if node $i$ is in the first layer.
81  We also have:

$$\sum_{i=1}^{n} \text{Dim}(\vec{w_i}) = N \text{ and } w \in \mathbb{R}^N$$

82  To damage a neural network, we define a damage operator $D_i$, that damages node $i$ in the network.

$$D_i : \mathbb{R}^N \longrightarrow \mathbb{R}^N$$

$$w' = D_i(w) \begin{cases} \vec{w'_i} = \mathbf{0}, \\ \vec{w'_j} = \vec{w_j} \end{cases}$$

83  To repair a neural network, we define a rescue operator $r_{\{i,j\}}$. Here $\{i, j\}$ refers to the set of damaged nodes. The rescue operator forces the
84  network to descend the loss manifold, while fixing nodes within the set and their connections to zero. Rescue of the network is achieved by
85  performing a constrained gradient descent on the networks' loss manifold.

$$r_{\{i,j\}} : \mathbb{R}^N \longrightarrow \mathbb{R}^N$$

$$w' = r_{\{i,j\}}(w) \begin{cases} \vec{w'_i} = \mathbf{0}, \vec{w'_j} = \mathbf{0}, \\ \vec{w'_k} = \vec{w_k} - \eta \frac{\partial L}{\partial \vec{w_k}} \end{cases}$$

86  where, $\eta$ is the gradient step-size and $\frac{\partial L}{\partial \vec{w_k}}$ is the gradient of the loss function of the neural network along $\vec{w_k}$

87  A damage-repair sequence involves the application of a damage operator followed by a repair operator.

$$w' = r_{\{i\}}(D_i(w))$$

88  A stochastic damage-repair sequence involves the random sampling of a damage operator from D, followed by the application of an
89  appropriate repair operator (ensuring that gradient descent is performed on remaining undamaged nodes).

$$w' = r_{\{i\}}(D_i(w)) \text{ where, } i \sim P(i) = \frac{1}{n}$$

90  We define a random variable $\mathbf{D}$ to sample an operator $D_i$ from the set of all possible damage operators $= \{D_i : i \in \{1, ..., n\}\}$. An iterative
91  damage-repair sequence is the repeated application of a random damage operator $\mathbf{D}$ coupled with a deterministic repair operator $r_{\{i,j,k,...\}}$,
92  that ensures all damaged nodes maintain a zero edge-weight, while other weights are plastic. Here, we show the long-hand and short-hand
93  notation for the iterative application of damage-repair operators.

$$w' = r_{\{i,j,k\}}(D_k(r_{\{i,j\}}(D_j(r_{\{i\}}(D_i(w)))))) \text{ where, } i, j, k \sim P(i, j, k) = \frac{1}{n^3}$$

$$w' = (r \circ \mathbf{D})^3(w)$$

94  We hypothesize that $\exists$ an open set of networks $U$, that constitutes an **invariant set**, where:

$$\text{if } w \in U, \text{ then}$$
$$(r \circ \mathbf{D})^m(w) \in U \ \ \forall m$$

We also claim that the invariant set $U$ is **path-connected**, ie given any two points from this topological space ($U$), there exists a path ($\gamma$) that connects the two points, starting at one point and ending at the other.

$$\text{For any two points, } w_1 \text{ and } w_2 \ \ \exists \gamma : [0,1] \longrightarrow U, \ \text{ such that:}$$
$$\gamma(0) = w_1, \gamma(1) = w_2 \text{ and } \gamma(t) \in U \ \ \forall t \in [0,1]$$

Our numerical results strongly suggests the presence of an invariant, path-connected topological space $U$ in the neural networks' loss manifold. In our experiments, the invariant, path-connected set is a collection of trained networks, whose image corresponding to the application of a damage and repair operator lies in the same set, visualized by the thick black arc (as shown in figure-4) obtained by tSNE embeddings of the high-dimensional network (w). We observe that iterative application of the damage-repair operator on a network sampled from $U$ results in a series of networks that belong to the same set $U$. This is observed in fig-4b & fig-4d. The red lines indicate damage of network, while the green lines correspond to repair of damaged networks. This hints at the possibility that $U$ is an invariant set. We also interpolated between all pairs of networks sampled from $U$ and observed that all the interpolated networks were present in $U$ as well.

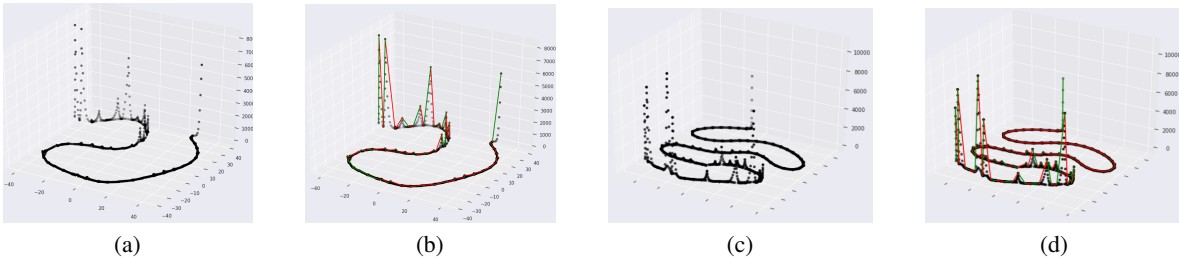

| (a) | (b) | (c) | (d) |

Figure 4: **Geometric features of the neural networks' loss landscape:** The x,y axes are tSNE embedding and z axes is the loss of the network (a,b) [tSNE] MLP with 1-hidden layer network (c,d) [tSNE] of MLP 2-hidden layers network (b,d) The green and red lines refer to repair and damage of networks respectively.

# 5 Discussion

In this paper, we address a pertinent question of how neural networks in the brain, or in engineered systems respond to damage of their units and whether there exists efficient strategies to repair damaged networks. We observe a phase transition behavior as we incrementally delete nodes from the neural network as the rate of decline of performance steeply increases after crossing a critical number of node deletions. We discover that damaged networks can be rescued and the iterative damage-rescue strategy produces networks that are highly resilient to perturbations, and can be rescued within a small number of training cycles. This is enabled by the putative presence of an invariant, path-connected set in the networks' loss manifold. Although we have shown numerical results that strongly suggest the presence of invariant sets in the loss manifold, our future work will focus on analytically proving the presence of these topological spaces in the loss manifold, through the formalization presented in the paper, and the use of the Koopman operator machinery, amongst others.

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
