# OpenReview forum: "Efficient rescue of damaged neural networks"
_NeurIPS.cc/2019/Workshop/Neuro_AI — Submitted to Real Neurons & Hidden Units @ NeurIPS 2019_

### Official Review · AnonReviewer3 · 2019-09-22
**A simple yet seemingly effective idea, exposed weakly with lots of clutter - feels very preliminary**

**Clarity:** 2

**Comment:**

As written, it is far from clear what the contributions of this submission are meant to be. The presented results range from providing very shallow to potentially deep insight, at best connected through future research to be done. As such, the work feels very preliminary in nature.

**Category:**

Common question to both AI & Neuro

**Clarity Comment:**

Color code is inconsistent across figures. A thorough description of the "iterative damage-repair paradigm" is never provided, leaving a lot for the reader to infer. Moreover, presentation of the performance of this algorithm (i.e. section 4) precedes the formal description of the algorithm, which reads awkwardly. The results presented in each section are never clearly related to each other.

**Evaluation:**

2: Poor

**Importance:**

3: Important

**Importance Comment:**

Although not explicitly discussed in the paper, it seems that an important hypothesis suggested by this work is that the constant rewiring observed in the brain might be a mechanism for building resilience to structural damage into neural circuits. From the perspective of AI, it is not clear that the kinds of damage considered here are actually a problem for hardware implementations of neural networks, limiting the importance or applicability of the proposed iterative damage-repair algorithm.

**Intersection:**

4: High

**Intersection Comment:**

Two important properties of biological neural networks are (1) their resiliance to damage and (2) the constant turnover and change of synaptic connections. This submission addresses how to build such properties into artificial neural networks using gradient descent. The results thus constitute a method to incorporate a property of biological neural networks into artificial neural networks. Unfortunately, few advantages to this are explored beyond resilience to a kind of damage unlikely to occur to artificial neural networks. Alternatively, one could interpret the results as providing a machine learning-inspired solution to understanding the nature of these biological properties. Unfortunately, this interpretation is hardly discussed or explored.

**Rigor Comment:**

The damage-repair algorithm proposed in this paper is principled and well demonstrated to achieve its goal (figure 3). However, because the algorithm is never clearly delineated, it is not clear how its data requirements compare to standard training schemes - are more epochs needed to train a network under the iterative damage-repair paradigm? Or can exactly the same amount of data be used as when training the network using standard SGD? If the former, then the interpretation of figure 3 changes drastically.

The significance of all the other results in this paper are far from clear. Figure 2 seems little more than trivial and irrelevant to the subsequent results. The significance of figure 4 is not well established, and moreover not discussed in the context of extensive previous literature on the existence of local minima in the loss landscape of deep networks. Moreover, it is not obvious that an alternative scenario to figure 4 could be possible, given that these networks differ structurally along a presumably relatively small set of gradient descent steps. Additionally, the non-linear dimensionality reduction could hide complications in this picture. It seems like some kind of control experiment is missing here to clarify the meaning of these results. Lastly, the nearly one page of mathematics mainly consists of definitions, without adding any rigor to the results or arguments ultimately made. That said, the authors do explicitly state in the discussion that the presented formalism may provide a basis for future research.

**Technical Rigor:**

2: Marginally convincing

---

### Official Review · AnonReviewer2 · 2019-09-22
**Poorly written and unconvincing, lacks neuroscience**

**Clarity:** 2

**Comment:**

I think the question of damage is of course relevant to both AI and neuroscience. I just don't think the authors have made a convincing case that their algorithm is inspired by any biological principle or could be taking place in the brain.

**Category:**

Common question to both AI & Neuro

**Clarity Comment:**

I do like the overall structure of the paper: first introducing damage, then simple repair, then the more complex damage-repair procedure. Yet the overall quality of the writing is mediocre. It is argued that a these networks are used for "cognitive tasks", but the only task evaluated here is the classical AI task of image recognition; cognitive tasks would be much broader in my opinion.

There is discussion of neuromorphic chips. I am not familiar with the details of these, but is node failure a common problem with them? It seems like more of an issue for biological networks, but I am not sure.

The figures are rather small with small labels.

The authors did not follow the formatting instructions since they changed the margin size. When I printed the paper for review the line numbers cut off.

Specific suggestions:
* The introduction has a good amount of awkward language and run-on sentences that could use editing.
* L. 23, "powerful paradigms" rephrase
* L. 36, what makes this "physical" damage?
* L. 39 "simple cognitive task" is a stretch
* L. 51, title your sections with a declarative statement rather than a question
* L. 62, write out "versus"

**Evaluation:**

1: Very poor

**Importance:**

1: Irrelevant

**Importance Comment:**

Finding neural networks that are robust to loss of neurons/connections is an important problem.  The results seem to be:
1) Removing nodes damages the network
2) Retraining can recover damaged networks
3) A damage-retrain procedure during training is better than doing repair later
However, this paper is not well-written, technically poor, and lacks the necessary tests/clarity to convince me especially about point 3. No attempt is made to connect to dropout, despite it being very similar.

**Intersection:**

1: Very low

**Intersection Comment:**

There is some discussion in the intro about how neurons in real networks may be damaged, but as far as I can tell, this is the only real connection to neuroscience. Everything else is just standard computational methods for artificial neural networks. I realize this is harsh, but I got the impression the authors just searched around for a few neuroscience citations to add in to a paper which is mostly a computational study.

The statement that in "most biological systems ... networks are constantly being perturbed and repaired" is debatable. I would say that yes neurons do die in the brain throughout our lifetime, but many of them stay around for a very long time, too. It probably depends on which part of the nervous system you are talking about. I think the authors could have conducted a more thorough literature review and found a lot of work on biological networks that study node removal, since I am aware of at least a few papers along those lines applied to traumatic brain injury.

However, if I were to be generous, I do think that biological systems probably apply principles like "learning under constant damage". But the mechanisms would be different and perhaps related to the stochasticity of neurons (wild speculation).  There is little evidence that real neurons can do backpropagation, even though it's a perennial topic at NeurIPS.

**Rigor Comment:**

The mathematics in this study are poorly laid out and not convincing. As is, the results would not be reproducible. The notation is sloppy and not well-described. It isn't really clear from the description exactly how the damage-repair procedure differs from repair strategy 1, "functional sub-network training". Both perform gradient descent on the non-damaged nodes; presumably the difference is how many gradient descent repair steps are made between damage steps.

I also take issue to the emphasis that damaging the network is a phase transition. I think anyone could expect that damaging nodes will make the performance decrease, whereas phase transitions refer to a variety of other phenomena (scaling, etc.) that the authors have not investigated. Basically, it makes a straightforward result sound cooler.

In the rescue sections (3 & 4), the number of repair cycles needed to rescue a network is plotted a few times. However, what constitutes "rescue" is not defined. I would guess it means # of epochs until performance is within some threshold of the performance before damage, but this should be stated explicitly. For the damage-repair strategy, I'm not sure we're seeing exactly the same thing plotted on the "Retrain cycles" axis in Fig 3. Do these correspond to the same number of iterations? How do these numbers compare to the initial number of cycles used to train the networks? Test error is never evaluated but could; everything is done with training error.

Finally, the whole argument about a damage-repair attractor does not convince me. The damage operation permanently removes a damaged neuron from the network. Therefore, the network gets smaller with each damage step. So the only attractor I can imagine is the eventual fixed point with no undamaged nodes and no network. This is a serious problem, unless I've misunderstood something crucial.

* I would suggest applying your technique to a state-of-the-art deep network rather than one you've trained yourself. That would be more convincing to the AI audience.
* Figure 1: The MLP with 1 layer actually does quite well until almost all of the nodes are removed... it is quite robust.
* L. 48 "critical threshold" is never defined. These don't look critical, since the phase transition is quite smooth down to removing all the nodes.
* L. 79, usually one would use a matrix W for this collection of weights; W is just one layer in the network but this isn't explained. How is N defined?
* L. 80, what is phi?
* L. 81, Dim is not defined, I think you mean number of nonzeros (i.e. the 0-norm)
* L. 81, w \in R^N is unclear; I would guess w is a d by n matrix with N nonzeros, but this is the wrong notation for that
* L. 82, missing an equals sign after D_i(w), the piecewise is also wrong, you mean w_j = 0 if j =1 and w_j = w_j otherwise
* L. 85, You never discuss picking a step size, or whether you use full gradient descent or stochastic gradient descent. These details are important.
* L. 85, again missing an equals sign before the piecewise definition, messed up similarly as before.
* L. 85, unclear why {i, j} is here, it'd be better to introduce some set S(t) for the damaged nodes at time t
* Ll. 89 & 93, the probabilities are wrong, since you are only picking from the remaining undamaged nodes (right?). Also, I don't think writing the composition operation really adds anything here.
* Figure 4: t-SNE can make things look nicer than they really are; are your results stable to other kinds of embeddings like isomap, etc?

**Technical Rigor:**

1: Not convincing

---

### Official Review · AnonReviewer1 · 2019-09-23
**Interesting idea for improving network resilience based on geometric understanding but could have been better executed**

**Clarity:** 2

**Comment:**

The idea of making AI systems more robust by continual damage and repair is certainly interesting, and the authors argue that it is feasible and effective. I also appreciate the attempt to connect this work to geometric features of the loss landscape. As written, however, the manuscript was rather hard to follow and did not seem very convincing. It also would have been nice to connect this idea to the much more well studied notion of dropout, as well as to discuss the effect of the damage-repair scheme to the network's response to adversarial inputs.

**Category:**

Neuro->AI

**Clarity Comment:**

As written, the use of damage and repair operators and the connection to topological spaces came across as a rather forceful attempt to use abstract mathematics. The result was unfortunately confusion rather than simplification. The use of unconvential notation (e.g. calling N the number of weights) also hindered understanding, and the network architecture was hard to understand, e.g. whether i referred to nodes or layers. Further, presenting long- and short-hand notation near line 93 seemed to serve no obvious purpose.

Figure 4 was also rather hard to parse.

Generally, I had to reread most paragraphs to understand what was being communicated.

**Evaluation:**

3: Good

**Importance:**

3: Important

**Importance Comment:**

The authors show how iteratively damaging and repairing networks produces more damage-robust networks and connect this to invariant parameter sets and connected paths through the loss landscape. This is a useful direction for making AI systems robust to physical damage and to understanding the geometry of the training loss landscapes. The idea to characterize damage and repair as operators on the weight matrix is also novel, and could maybe be worth further investigation.

**Intersection:**

2: Low

**Intersection Comment:**

The authors motivate resilience to damage based on biological considerations, but that is the only neuroscience connection.

**Rigor Comment:**

While suggesting the potential for rigor, the authors' attempt to characterize damage and repair as operators on network parameters did not end up making the approach feel more principled. The authors would have been better off exchanging some of the more advanced mathematical ideas (which did not seem to add to their arguments) for more detail about e.g. the loss function and specific properties of the repair operator. Further, the authors compute network performance only in terms of training accuracy, raising doubts as to how well damage and repair affect performance on test data.

One thing I also did not understand was how a network that receives continual damage by node deletion would remain robust to damage, since eventually there would be no more nodes remaining.

The idea of connected paths through the loss landscape was also interesting but not properly detailed.

**Technical Rigor:**

2: Marginally convincing

---

### Decision · Program_Chairs · 2019-10-01

**Decision:**

Reject

**Comment:**

Unfortunately, we had more submissions than we could accept and based on the review process, we have decided not to accept your submission.  Nevertheless, thank you for your submission and interest in our workshop.